# Investigation of Imidazolinone Herbicide Resistance Gene with KASP Markers for *Japonica*/*Geng* Rice Varieties in the Huanghuaihai Region of China

**DOI:** 10.3390/plants13081097

**Published:** 2024-04-14

**Authors:** Peng Liu, Wenjie Feng, Tao Wang, Huadong Zhang, Shuaige Mao, Hua Zhang, Wenchao Huang, Haifeng Liu, Shangzong Feng, Zhaohui Chu

**Affiliations:** 1State Key Laboratory of Hybrid Rice, Hongshan Laboratory, College of Life Science, Wuhan University, Wuhan 430072, China; lp2333@whu.edu.cn (P.L.); 2019202040069@whu.edu.cn (T.W.); zh13854919696@163.com (H.Z.); 2021202040076@whu.edu.cn (S.M.); wenchaoh@whu.edu.cn (W.H.); 2Jining Academy of Agricultural Sciences, Jining 272031, China; jnnky_fwj@126.com; 3Tancheng Jinghua Seed Co., Ltd., Linyi 276100, China; zhd202071599@163.com; 4College of Agronomy, Shandong Agricultural University, Taian 271018, China; hliu1987@sdau.edu.cn; 5Agro-Technical Popularization Centre of Linyi City, Linyi 276000, China

**Keywords:** rice, imidazolinone, OsALS1, herbicide-resistant, KASP, InDel, resequencing

## Abstract

Rice is a staple food for more than half of the global population due to its food security and sustainable development. Weeds compete with crops for sunlight and indispensable nutrients, affecting the yield and quality of crops. Breeding herbicide-tolerant rice varieties paired with herbicide application is expected to help with weed control. In this study, 194 *Japonica*/*Geng* rice varieties or lines collected from the Huanghuaihai region of China were screened by Kompetitive Allele-Specific PCR (KASP) markers based on four mutation sites within *OsALS1* (*LOC_Os02g30630*), which is the target of imidazolinone (IMI) herbicides. Only the *OsALS1^627N^* haplotype was identified in 18 varieties, including the previously reported Jingeng818 (JG818), and its herbicide resistance was validated by treatment with three IMIs. To investigate the origin of the *OsALS1^627N^* haplotype in the identified varieties, six codominant PCR-based markers tightly linked with *OsALS1* were developed. PCR analysis revealed that the other 17 IMI-tolerant varieties were derived from JG818. We randomly selected three IMI-tolerant varieties for comparative whole-genome resequencing with known receptor parent varieties. Sequence alignment revealed that more loci from JG818 have been introduced into IMI-tolerant varieties. However, all three IMI-tolerant varieties carried clustered third type single nucleotide polymorphism (SNP) sites from unknown parents, indicating that these varieties were not directly derived from JG818, whereas those from different intermediate improved lines were crossed with JG818. Overall, we found that only *OsALS1^627N^* from JG818 has been broadly introduced into the Huanghuaihai region of China. Additionally, the 17 identified IMI-tolerant varieties provide alternative opportunities for improving such varieties along with other good traits.

## 1. Introduction

It is estimated that the world population will increase to over 9 billion by 2050 and determining how to feed this population is a serious challenge [1]. Rice (*Oryza sativa* L.), an important cereal crop, is the staple food for more than half of the global population, providing 21% of the world’s calories [2,3,4]. Maintaining and increasing rice production is crucial for food security. Weeds are a serious biotic stress that pose a threat to rice yield, aggressively compete with crops for nutrients, light and other important resources, leading to greater than 40% yield loss in some regions [5,6,7]. The shortage of labor in rice-planting areas is worsening because of the large number of people flowing to cities with the acceleration of the urbanization process, especially in China. To address this issue, rice cultivation methods are gradually shifting towards direct-seeded rice (DSR), and chemical herbicides are preferred for broad-spectrum weed control due to their low cost and high efficiency [8]. However, improper use of herbicides usually results in indiscriminate hindrance to crop growth and ultimately leads to a decrease in crop yield. Additionally, breeding and planting herbicide-tolerant crops are highly important for easing weed infestations and maintaining sustainable crop production. Researchers have obtained herbicide-resistant rice through modern methods, such as transgenic technology, mutagenesis, breeding and gene editing, but restrictions on genetically modified plants make them difficult to plant and commercialize globally [9,10,11,12]. Therefore, the genetic improvement of cultivated rice via natural selection or non-transgenic mutagenesis can ameliorate its tolerance to certain herbicides, along with the safe application of such herbicides to selectively kill weeds [8].

Plant resistance or tolerance to herbicides mainly comes from target site resistance (TSR) caused by mutations in a single gene, and most herbicides are designed to target specific enzymes or proteins. The acetyl-lactate synthase (ALS) protein is involved in the first synthesis of branched-chain amino acids (valine, leucine and isoleucine), which are vital for protein synthesis and plant growth. ALS inhibitors are Group B herbicides and are favored because of their broad spectrum, extreme activity, strong selectivity and high biosafety [13]. To date, more than 50 commercial ALS inhibitors have been developed for weed control and essential crop protection [13,14]. They include five main families of compounds with different structures: imidazolinones (IMIs), sulfonylureas (SUs), triazolo-pyrimidines (TPs), pyrimidine salicylic acids (PBs) and sulfonyl-aminocarbonyl-triazolinones (SCTs) [13]. IMIs, including imazameth, imazamox, and imazethapyr, are among the most widely utilized groups of inhibitors of ALS [13]. Moreover, mutations at specific sites within the ALS coding region have been verified as increasing crop and weed tolerance to IMIs [12,15,16,17,18]. The amino acid substitution resulting from the base mutation generates a novel ALS isoform that is less sensitive to IMIs but does not affect its enzyme activity compared with that of the wild type [19]. Currently, mutations at four amino acid sites, A179V [20], W548M [21], S627N [22] and G628E [12], have been reported to confer IMI resistance in rice. Some of these plants have been widely planted in paddies [12].

Advances in high-throughput sequencing technology have spawned plentiful single nucleotide polymorphism (SNP) and insertion-deletion (InDel) sites in many crops, promoting the development of genotyping arrays based on these variants [23]. The abundance of tools has accelerated the pace of molecular genomic breeding. Kompetitive Allele-Specific PCR (KASP) is a homogeneous, fluorescence-based genotyping technology based on allele-specific oligo extension and fluorescence resonance energy transfer (FRET) for signal generation [24]. Two upstream primers in the KASP reaction system that differ in only 3′ terminal bases have the ability to discriminate between different haplotypes [24]. Due to its advantages of high throughput, low cost, high reproducibility and easy operation, KASP has great application potential in the fields of agronomic trait improvement, variation screening and map-based cloning [25,26].

The annual planting area of *Japonica*/*Geng* rice in the Huanghuaihai region is approximately 1.73 million hectares, accounting for 18.6% of the rice planting area in China. The abundant sunshine, flat land and intensive and meticulous farming habits of locals make it one of the primary regions for high-quality rice production in China [27]. However, *Gramineae* weeds, including *Echinochloa crusgalli* (L.) *P. Beauv*., *Oryza sativa Linnaeus*, *Alopecurus aequalis*, etc., have become serious biotic stresses that have plagued the rice-wheat rotation agricultural model and yield in this region [28]. The adoption of light and simple cultivation and DSR technology has experienced rapid development and widespread promotion in this region due to its water and labor-saving benefits. IMI-tolerant rice varieties that originated from mutagenesis by ethyl methyl sulfonate (EMS) or the cultivar Jingeng818 (JG818) have been introduced into this region to cooperate with the cultivation system [29]. Therefore, screening and breeding herbicide-tolerant varieties are of prime importance for the sustainable development of rice in this region.

In this study, we selected IMI-targeted *OsALS1* as the research object and screened 194 collected *Japonica*/*Geng* varieties with four KASP markers. Eighteen *OsALS1^627N^* haplotype varieties were identified to be tolerant to IMI and originated from the same donor, JG818, based on PCR analysis with six developed InDel markers. Resequencing and sequence analysis were performed on three IMI-tolerant varieties and one of their parent plants.

## 2. Results

### 2.1. Development of Specific KASP Markers to Screen for Each Variant of ALS1^179V/548M/627N/628E^ from Collected Japonica/Geng Rice Varieties or Lines

To investigate the IMI-tolerant rice varieties in the Huanghuaihai region of China, we focused on four sites (*ALS1^179V^*, *ALS1^548M^*, *ALS1^627N^*, and *ALS1^628E^*) that acquired IMI herbicide resistance through mutagenesis breeding, as previously reported [12,20,21,22]. Four specific KASP markers were developed for 194 *Japonica*/*Geng* rice varieties or lines collected from the Huanghuaihai region (Figure 1). Only the *OsALS1^627N^* haplotype was identified for 18 varieties: JG818, Runnong11v2, Jifeng105, Jindi672, Huageng5v2, Runnong303, Huaidao678, Runnong17, Suyuannuo879, Runnong3, Hainan5, Yangchan1013, Habo1699, Runnong5818, 63646, Runnong802, Kuo32, and Jindao565 (Figure 1c, Appendix A). Among these varieties, only JG818 was validated to be an *OsALS1^627N^* haplotype from more than 7000 rice lines through large-scale screening of IMI herbicide treatment [29]. Overall, in our collected rice populations, approximately 9.28% of the total varieties contained the *OsALS1^627N^* haplotype.

### 2.2. Imidazolinone Herbicide Resistance Assay for OsALS1^627N^ Rice Varieties

As JG818 contains the *OsALS1^627N^* mutation, to validate JG818 in our collections, we cloned *OsALS1* and sequenced from JG818 as well as from the other three *Japonica*/*Geng* rice varieties, Nipponbare, Zhonghua11 and Kenxiang48, and two *Indica*/*Xian* rice varieties, 93-11 and Minghui63 (Appendix A). Sequence alignment revealed that only one specific nucleotide difference (1880 G to A) existed between JG818 and the other five varieties. This mutation caused an amino acid substitution from serine to asparagine at position 627, which were named *OsALS1^627S^* and *OsALS1^627N^*, respectively. Another mutant site (1927 G to A) is jointly owned by JG818 and Minghui63, which cause the amino acid to change from valine to methionine at position 643; these are referred to as *OsALS1^643V^* and *OsALS1^643M^*, respectively. To probe whether the two substitution sites affect tolerance to IMIs, a phenotype validation experiment on six varieties was performed, and only JG818 was completely resistant to imazethapyr (Figure 2a). Moreover, Minghui63, which carries another variation type, *OsALS1^643M^*, showed weak resistance to IMIs (Figure 2a). Taken together, the JG818 strains from the collections carried *OsALS1^627N^*, which confers IMI tolerance.

To further evaluate the impact of changes in amino acids at the *OsALS1^627N^* site in rice, 21 rice varieties, including 6 *OsALS1^627N^* and 15 *OsALS1^627S^* haplotypes, were randomly selected from 194 rice populations for the IMI tolerance assay. All six *OsALS1^627N^* rice haplotypes showed complete resistance to all three commercial IMI herbicides, including imazameth, imazamox and imazethapyr, while the other 15 *OsALS1^627S^* varieties were sensitive and completely withered (Figure 2b). These results indicate that rice varieties harboring the *OsALS1^627N^* variant identified from KASP are resistant to IMI herbicides.

### 2.3. Development of Molecular Markers Tightly Linked to OsALS1^627N^ from Sequenced JG818

JG818 was the first IMI-tolerant rice variety introduced into the Huanghuaihai region. The other 17 KASP-positive varieties exhibited *OsALS1^627N^* variation, which is reminiscent of the same origin as that of the donor JG818. To confirm this speculation, Illumina second-generation sequencing was performed on JG818. Subsequently, the sequence was aligned with the reference Nipponbare genome sequence to determine the linkage and flanking of InDel markers with *OsALS1*. We developed 5 InDel and one cleaved amplified polymorphic sequence (CAPS) marker flanking *OsALS1* (Figure 3a, Table 1). Electrophoresis revealed that the PCR products of markers M1~M6 were clearly different between JG818 and Nipponbare (Figure 3b).

After validating the polymorphism between JG818 and Nipponbare, six markers were subjected to PCR analysis for all 17 *OsALS1^627N^* varieties and 15 randomly selected *OsALS1^627S^* varieties. As shown in Figure 3c, for all 17 positive varieties, JG818-specific DNA bands were amplified, whereas for 15 *OsALS1^627S^* varieties, Nipponbare-specific DNA bands were amplified for all six markers (M1 to M6). It contains approximately 598 Kb flanking the *OsALS1* region and ranges from M4 to M5 (Figure 3a). These results indicate that the above 17 *OsALS1^627N^* varieties share the same *OsALS1* locus originating from JG818.

### 2.4. Comparative Analysis of the Genome Sequence of Herbicide-Resistant Rice

To further confirm whether JG818 was the donor of the other *OsALS1^627N^* varieties, we selected Runnong11 and Runnong11v2 (IMI-tolerant improved line of Runnong11), Huageng5 and Huageng5v2 (IMI-tolerant improved line of Huageng5) and Jifeng105 (one parent of which is Huageng5) for IMI tolerance tests and whole-genome sequencing. Runnong11 and Huageng5 carry the *OsALS1^627S^* genotype, while Runnong11v2, Huageng5v2 and Jifeng105 carry the *OsALS1^627N^* genotype. Consistent with their genotypes, Runnong11v2, Huageng5v2 and Jifeng105 were tolerant to imazethapyr treatment, whereas Runnong11 and Huageng5 were sensitive (Figure 4).

After resequencing, 4.09 to 10.18 Gb of mapped reads were generated that covered 96.85% to 97.55% of the rice genome for all six rice varieties or lines (Table 2). Through comparative analysis of the genome sequences of Runnong11v2 with those of JG818 and Runnong11, we obtained 344,551 SNPs distributed evenly across 12 chromosomes. Among the total SNPs, 59,163 SNPs (17.17% of the total) were identified for JG818, while 258,674 SNPs were identified for Runnong11 (Table 2). In addition to the enrichment of SNPs from JG818 in the OsALS1 locus in the chromosome 2 (Chr) cluster, we also found that JG818-specific SNPs were clustered in the centromeric region of Chr 6, the short arms of Chr 5 and Chr 12, and the long arms of Chr 1, 3 and 7 (Figure 5a). In particular, 25384 SNPs (7.37% of the total) were not detected in either JG818 or Runnong11 (Figure 5a). Similar findings were also identified for the other two groups. According to the resequencing analysis of Huageng5, Huageng5v2 and JG818, 347,376 SNPs were evenly distributed on 12 rice chromosomes. Among them, 280,909 SNPs were consistent with Huageng5. Approximately 44,277 SNPs represented 12.75% of the total number of JG818-specific SNPs and were enriched on the long arms of Chr 1, 5, 7, 8, 9, and 12, the short arms of Chr 4, 6, 7 and 11, and the OsALS1 locus on Chr 2 (Figure 5b). Furthermore, 20,609 SNPs, which represented 5.93% of the total SNPs, were not identified in either JG818 or Huageng5 (Figure 5b). Additionally, the rice line Jifeng105 was also analysed and aligned with JG818 and Huageng5. A total of 391,478 SNPs were identified on all 12 chromosomes. There were 289,763 SNPs associated with Huageng5 and 79,139 SNPs (20.21% of the total) associated with JG818 (Figure 5c). Furthermore, there were also 20,510 SNPs identified in neither JG818 nor Huageng5 (Figure 5c). From the cluster enrichment of JG818-specific SNPs, in addition to the OsALS1 locus, Chr 1, 6, 7, 8, 9, 11, and 12 contained 1 to 3 enriched regions highlighted in green (Figure 5c).

Overall, we found that there were long fragments of most chromosomes consistent with JG818 in all three *OsALS1^627N^* genotype rice genomes. In particular, three improved lines (Runnong11v2, Huageng5v2, and Jifeng105) flanking the *OsALS1* locus were 5–10 Mb in length on chromosome 2. In addition to this locus, many other clustered JG818-specific SNPs, represented by JG818-containing DNA fragments, have been identified on several other chromosomes. These results further suggested that *OsALS1^627N^* in the Huanghuaihai region originated from the same donor. It has been employed in the genetic improvement of *Japonica*/*Geng* rice.

## 3. Discussion

To feed up to 1.4 billion people, light and simple cultivation and DSR technology prevailed in the Huanghuaihai region, where the *Japonica*/*Geng* rice variety was mostly planted. Herbicide-tolerant *Japonica*/*Geng* rice varieties are closely linked with the cultivation system to be introduced and developed in this region. JG818 was the first non-transgenic IMI-tolerant *Japonica*/*Geng* rice variety identified from large-scale screening of more than 7000 varieties with IMI treatment [29]. After the approval certificate, it was introduced into this region in 2015. In this study, we screened 17 other IMI-tolerant varieties or lines other than JG818 from 194 collections (Figure 1). The distribution frequency was approximately 9.28% (Appendix A). All the other 17 IMI-tolerant varieties or lines were identified to be the same *OsALS1^627N^* haplotype varieties derived from JG818 (Figure 3). There are still many improved IMI-tolerant varieties derived from crosses with JG818 or its improved lines that have not been listed in these collections in this region [22,30,31]. In summary, the herbicide gene derived from JG818 has received much attention and has been favored and has spread to local breeders and farmers over the past nine years.

Changes in the key amino acids in *OsALS1* can reduce the sensitivity of the target enzyme to the corresponding herbicide. The target site resistance of ALS involves one or more specific amino acid mutations, namely, Ala122, Pro197, Ala205, Asp376, Trp574, Ser653 and Gly654 (corresponding sites in *Arabidopsis thaliana*) [32]. The resistance type and effect of herbicides were determined by the amino acid mutation site and type of ALS. For example, ALS1^122Y/V^ showed high resistance to SU and TP herbicides but low resistance to IMI herbicides, while ALS1^122T^ amino acid mutations showed high resistance only to IMI herbicides [33,34,35,36,37]. Since the first IMI-tolerant rice was marketed in the USA in 2001, it was developed from the EMS mutagenesis line AS3510 [38]. To date, a series of ALS inhibitor-resistant mutation sites containing amino acid mutations at Ala179, Trp548, Ser627, and Gly628 have been found in cultivated rice [12,20,21,22,39]. The long-term use of a single herbicide resistance gene, particularly a haplotype mutation, will easily increase the risk of generating herbicide-resistant weeds. To our surprise, only one variation, *OsALS1^627N^*, was identified from a total of 194 collections (Figure 1). These findings suggest that donors containing different *OsALS1* haplotypes should be used to improve IMI-tolerant rice in the Huanghuaihai region in the future. Recently, some rice strains with mutations conferring tolerance to acetyl-CoA carboxylase (ACCase) inhibitor herbicides and glufosinate have been generated via EMS mutagenesis or heavy ion beam treatment in China [40,41,42]. The introduction of IMI-tolerant varieties into the Huanghuaihai region and rotational planting of IMI-tolerant varieties are important for sustainable weed control.

Various single-marker methods, including allele-specific PCR (AS-PCR), CAPS, temperature-switch PCR (TS-PCR), ultrahigh-throughput chips and gene resequencing, have been developed for SNP genotyping. KASP is a fluorescence-based homogeneous genotyping technique originally developed by KBioscience in the United Kingdom [24]. KASP can be used for accurate biallelic genotyping of target SNPs due to its accuracy, specificity, flexibility, low cost and high efficiency [43]. Additionally, the combination of multiple high-throughput sequences and KASP can compensate for the shortcomings of both methods and plays an important role in germplasm resource identification, marker-assisted breeding, genetic map construction and QTL mapping [33,44,45]. In this study, we developed four KASP markers based on *OsALS1* variations, which facilitated the screening of *Japonica*/*Geng* rice in the Huanghuaihai region. Additionally, PCR-based marker linkage to *OsALS1* is a convenient method for marker-assisted selection of IMI-tolerant rice [46]. We developed six PCR-based markers linked to *OsALS1* based on comparative analysis of JG818 resequencing data that will facilitate marker-assisted selection of IMI-tolerant rice (Figure 3). In particular, these markers showed no polymorphisms among 15 randomly selected IMI-sensitive varieties or lines (Figure 3c). These markers could be used instead of KASP markers to screen for *OsALS1* variation in JG818.

Previously, there have been many attempts to improve the IMI herbicide resistance of *Japonica*/*Geng* variety donors from JG818 without altering biotic stress resistance or agronomic traits [29,30,31]. Additionally, many IMI-tolerant rice strains have been generated via EMS mutagenesis [40,42]. However, the genetic backgrounds of these varieties have not been investigated. Here, three improved IMI-tolerant varieties or lines were resequenced and compared with their receptor parents and JG818. Unexpectedly, clustered JG818-specific SNPs were identified on most other chromosomes in addition to the *OsALS1* locus of Chr 2 (Figure 5). These results further supported that JG818 was the original donor of *OsALS1^627N^* for the 17 IMI-tolerant rice varieties or lines. Moreover, the presence of clustered unknown SNPs was detected in Runnong11v2, Huageng5v2 and Jifeng105, suggesting that they were directly derived from improved intermediate lines crossed between JG818 and an unknown receptor parent. These improved intermediate lines may be hidden in other identified IMI-tolerant varieties or lines without sequences or in those developed but not in our collections. Overall, we concluded that IMI-tolerant varieties or lines carrying only the *OsALS1^627N^* variation are broadly prevalent in *Japonica*/*Geng* rice in the Huanghuaihai region of China. 

## 4. Materials and Methods

### 4.1. Plant Material and Growth Conditions

The *Japonica*/*Geng* rice varieties Nipponbare and Zhonghua11 and the *Indica*/*Xian* rice varieties 93-11 and Minghui 63 used in this study were stored in our laboratory. The 194 Huanghuaihai varieties or lines were collected by Hua Zhang (Tancheng Jinghua Co., Ltd., Tancheng, China) and Wenjie Feng (Jining Academy of Agricultural Sciences, Jining, China). All seeds collected from 2021 to 2023 in the Huanghuaihai region were approved varieties or regional test lines, as listed in Appendix A. The seeds were presoaked with 75% ethanol to sterilize them and then sprouted in small pots (7.5 × 7.5 × 7 cm) containing a matrix of organic matter and vermiculite (mixed at a ratio of 2:1). All plants were cultivated in a phytotron of a plant growth breeding system (PGBS, Wuhan Greenfafa Institute of Novel Genechip R and D Co., Ltd., Wuhan, China) greenhouse with 12 h of light and 12 h of darkness, and the humidity was maintained at 70%.

### 4.2. Phenotyping of Imidazolinone Herbicide Resistance

Two- to three-week-old rice plants were sprayed with 5% imazethapyr (commercially available as Mucaojing^®^, Shandong CYNDA, Jinan, China), 4% imazamox (commercially available as Shengjin^®^, Jiangsu Flagchem, Nanjing, China), or imazameth (commercially available as Huolongjuan^®^, Shandong Kangqiao, Binzhou, China; 240 g/L). Phenotypes were observed and photographed two weeks later. Seedlings are insensitive to IMIs and grow normally after treatment. The presence of withered plants indicates that they are sensitive to IMIs.

### 4.3. DNA Extraction, InDel Marker Development and PCR

Genomic DNA was isolated from young leaves of each variety or line using a Plant Genomic DNA Kit (CWBIO, Beijing, China) following the manufacturer’s instructions. Sequences of the *OsALS1* locus were downloaded from the National Center for Biotechnology Information (NCBI) under accession numbers AP014958.1 for Nipponbare (18236120 to 18238054), CM025512.1 for Zhonghua11 (14461361 to 14463295), CM012054.1 for 93-11 (19848411 to 19850345), and CM003886.1 for Minghui63 (6574494 to 6576428). Primers of the M1 to M6 target sequences were designed, and their quality was assessed by PrimerPremier6 and Primer-BLAST (https://www.ncbi.nlm.nih.gov/tools/primer-blast/, accessed on 19 March 2024). The primer sequences were synthesized by Sangon Biotech (Shanghai, China). The 20 μL PCR mixture consisted of 10 ng of genomic DNA, 0.8 μM of each primer, 10 μL of 2 × Es Taq Master Mix (CW0718L, CWBIO, Taizhou, China) and ddH_2_O. The Mixture reacted in the Arhat 96 Thermal Cycler (Monad, Suzhou, China). The thermal cycling procedure was set as follows: pre-denaturation at 95 °C for 5 min, followed by 32~35 cycles of 95 °C for 15 s, 52~55 °C for 15 s, and 72 °C for 15 s. The PCR products were digested with *Hin*f I (FD0804, Thermo Fisher, Scientific, Inc., Waltham, MA, USA) and electro-phoretically separated on a 2.5% agarose gel at 110 V for 40 min, and the resulting products were imaged with a Tanon 2500 gel imager system (Tanon, Shanghai, China).

### 4.4. Whole-Genomic Sequencing and Data Analysis

Library construction and resequencing of the whole rice genome were performed by GENOSEQ Technology Ltd. (Wuhan, China). We used FastQC to conduct quality tests of the raw paired-end reads (https://www.bioinformatics.babraham.ac.uk/projects/fastqc/, accessed on 19 March 2024) and then used fastp (v0.21.0) to remove undesired reads [47]. The cleaned reads were aligned to the rice reference genome MSU7.0 (http://rice.uga.edu/, accessed on 19 March 2024) using BWA-MEM (v0.7.17-r1188) with default parameters [48]. The mapped results were sorted by SAMtools (v1.11) [49], and the PCR-duplicated mapping reads were marked by the MarkDuplicates program in GATK (https://github.com/broadinstitute/gatk, v4.0.2, accessed on 19 March 2024). Based on uniquely mapped reads, the HaplotypeCaller program in GATK was used to call germline SNPs. Subsequently, the raw SNPs were preliminarily filtered by the VariantFiltration program in GATK with the following parameters: --filter-expression “QD < 2.0 || MQ < 40.0 || FS > 60.0 || SOR > 3.0 || MQRankSum < −12.5 || ReadPosRankSum < −8.0” --filter-name ‘SNP_filter’.

### 4.5. KASP Marker Development and Genotyping Assay

The *OsALS1* sequences of Nipponbare, 93-11, and Minghui 63 were used as DNA templates. Primers were designed to amplify the allele sequence from JG818, ZH11, and Kenxiang48 (Table 3). The selection of mutation sites was based on alignment to the rice reference sequence and annotation of variations in A179V, W548M, S627N and G628E [12,20,21,22]. The KASP primers were designed and supplemented with the anchored sequence according to the instructions of the 2x Master Mix for ASPCR V1 (HC Scientific, Chengdu, China). The 5 μL KASP mixture consisted of 50 ng genomic DNA, 0.05 μM each upstream primer, 0.15 μM reverse primer, and 2.5 μL 2x Master Mix for ASPCR V1 (HC Scientific). KASP marker assessment was performed via Archimed X4 real-time fluorescent quantitative PCR system (Rocgene, Beijing, China) using the following thermal cycling program: pre-denaturing at 95 °C for 10 min; followed by 10 cycles of 95 °C for 20 s and 61~55 °C (gradient decrease of 0.6 °C per cycle) for 40 s; followed by 30 cycles of 95 °C for 20 s and 55 °C for 40 s; and the fluorescence signals of FAM and HEM were collected during each program. The relative values of fluorescence intensity were exported for processing and genotyping generation by Microsoft Excel 2019 (Microsoft, Redmond, WA, USA).

## 5. Conclusions

Based on the assessment of *OsALS1*-specific KASP markers, we identified 17 other rice varieties carrying the unique *OsALS1^627N^* haplotype in the Huanghuaihai region of China, which provides important evidence that naturally mutated *OsALS1^627N^* has been broadly introduced into cultivated *Japonica*/*Geng* rice varieties. These strains could serve as alternative donors to JG818 for improving IMI tolerance as well as improving other good agronomic traits. Additionally, it was first identified that all 17 IMI-tolerant varieties originated from the unique donor JG818.

## Figures and Tables

**Figure 1 plants-13-01097-f001:**
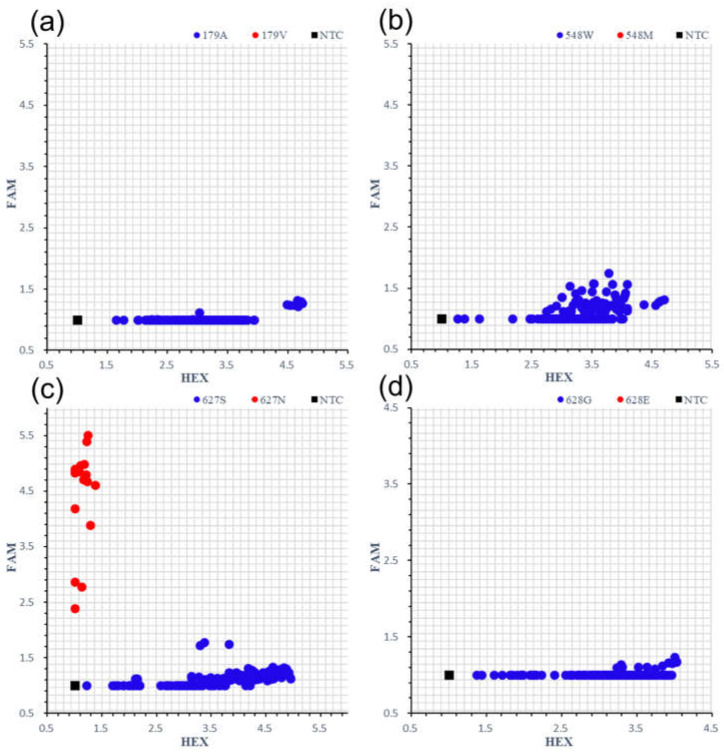
Genotyping of four KASP markers for 194 *Japonica*/*Geng* rice varieties or lines. A179V (**a**); W548M (**b**); S627N (**c**); G628E (**d**). The ratio of the fluorescence intensity of HEX and FAM to ROX was utilized for genotype analysis.

**Figure 2 plants-13-01097-f002:**
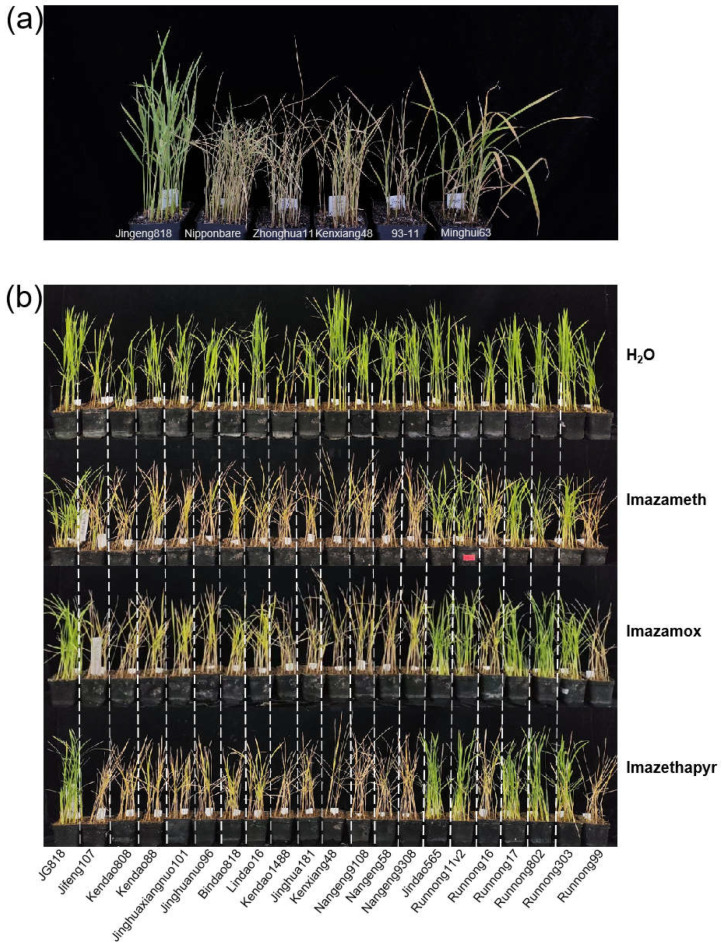
Phenotyping of imidazolinone herbicide tolerance. (**a**) Phenotypes of Jingeng818, Nipponbare, Zhonghua11, 93-11, and Minghui63 plants under imazethapyr treatment. (**b**) Phenotyping of 21 rice varieties or lines, including 6 *OsALS1^627N^* and 15 *OsALS1^627S^* haplotypes randomly selected from 194 accessions for IMI tolerance experiments with imazameth, imazamox and imazethapyr. Water was used as a control.

**Figure 3 plants-13-01097-f003:**
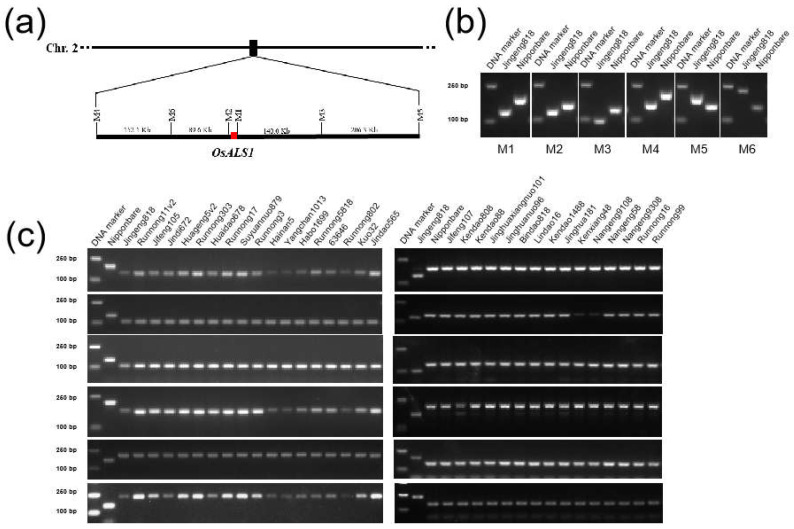
Development of PCR-based markers for linkage analysis of 18 *OsALS1^627N^* varieties. (**a**) Genetic location of the six developed markers on chromosome 2. (**b**) Polymorphism identified between Jingeng818 and Nipponbare. (**c**) Genotyping of 18 *OsALS1^627N^* and 15 randomly selected IMI-sensitive rice varieties or lines. Electrophoresis was performed on a 2.5% agarose gel with a 2 Kb DNA ladder.

**Figure 4 plants-13-01097-f004:**
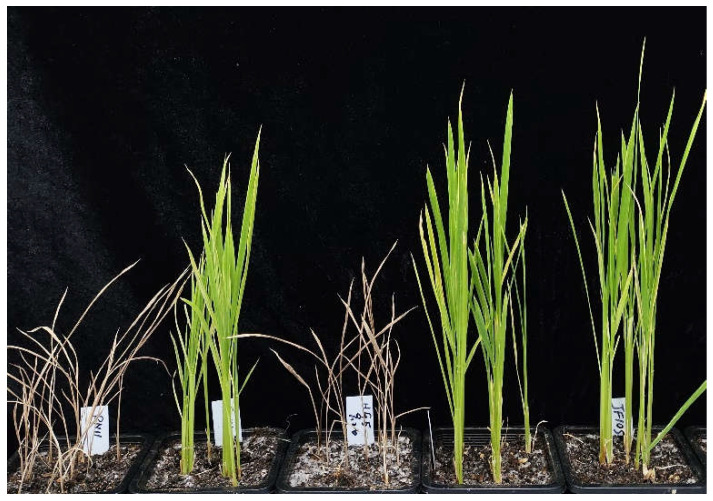
Phenotyping of Runnong11, Runnong11v2, Huageng5, Huageng5v2, and Jifeng105 under imazethapyr treatment. Images were taken 2 weeks after herbicide treatment.

**Figure 5 plants-13-01097-f005:**
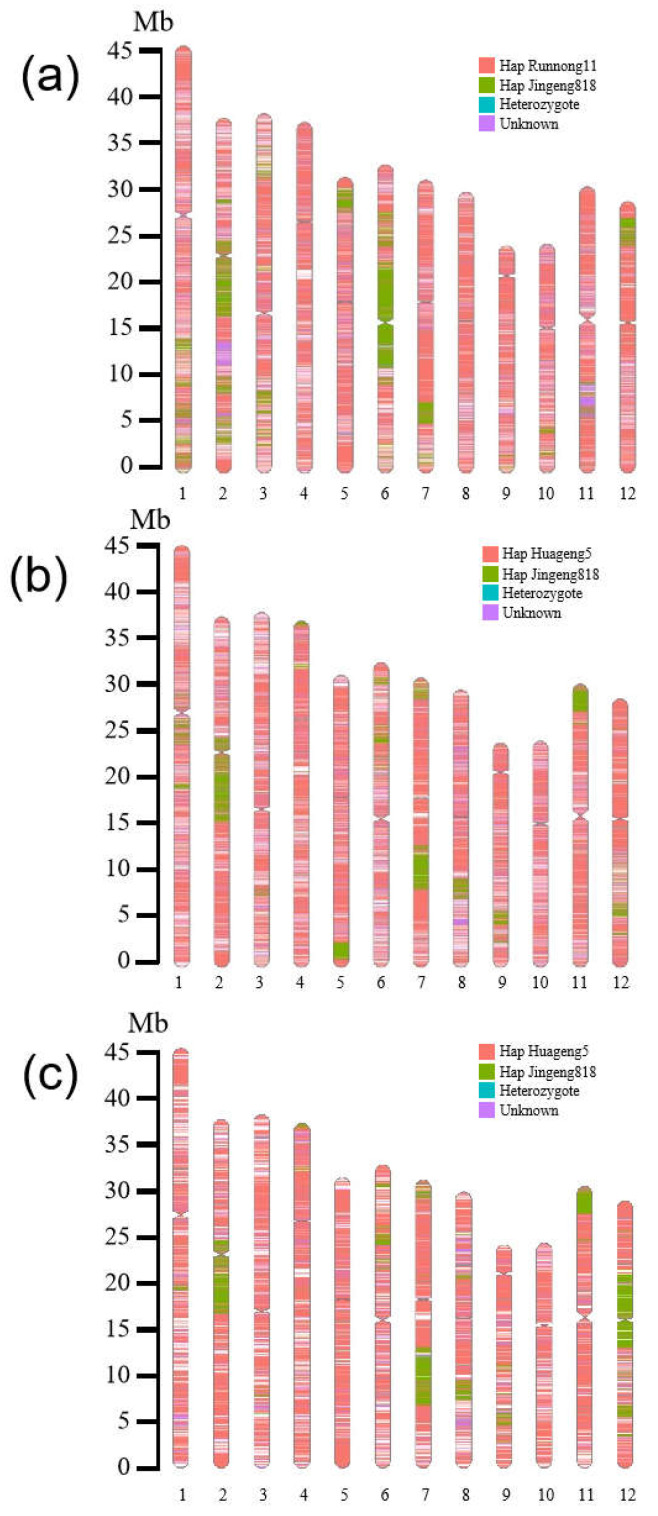
Comparative analysis of Jingeng818-specific SNPs based on genome sequence. (**a**) Runnong11v2 vs. Runnong11. (**b**) Huageng5v2 vs. Huageng. (**c**) Jifeng105 vs. Huageng5. The specific SNPs labelled in different colours are highlighted in the figures.

**Table 1 plants-13-01097-t001:** Primer sequence of six markers closely linked to *OsALS1*.

Marker	Forward Primer Sequence (5′-3′)	Reverse Primer Sequence (5′-3′)	Product Length (bp)
M1	CCAAAGGTTCCCTAATCAAGTT	CGGGTTTAGATGTGGGTTTC	142 ^1^/191 ^2^
M2	GGGAGTGATTCTGAATGTTA	AGATCCCAGATCATCCTTTC	134 ^1^/158 ^2^
M3	ACCATCGTCATCGCTGTC	CGCTCAGGAGATCGAAGAT	97 ^1^/138 ^2^
M4	TGTATGATAGTCAGCCTTGTG	GAGGTGGTAATAACGTGTGA	170 ^1^/219 ^2^
M5	ACACAGTGTGAACTAACCTA	CTAAACAGGGCCGTAGTATA	108 ^1^/154 ^2^
M6 *	ATCAGCATCCTCAGTTCCATAA	GCACAAAGTGAAAGACACATTC	220 ^1^/146 ^2^

* PCR product digested by *Hin*f I; Size of DNA fragment for ^1^ JG818 and ^2^ Nipponbare.

**Table 2 plants-13-01097-t002:** Re-sequencing data and mining SNPs for six rice varieties or lines.

Varieties/Lines	Clean Reads (Gb)	Mapped Reads (Gb)	Coverage	Phenotype *	Number of SNPs				
					JG818	Sensitive Parent	Heterozygous	Unknown	Total
Jingeng818	16.51	10.18	97.55%	T	-	-	-	-	-
Runnong11	6.39	6.34	97.41%	S	-	-	-	-	-
Runnong11v2	4.12	4.09	96.85%	T	59,163	258,674	1330	25,384	344,551
Huageng5	5.72	5.68	97.53%	S	-	-	-	-	-
Huageng5v2	5.25	5.23	97.33%	T	44,277	280,909	1494	20,696	347,376
Jifeng105	6.35	6.32	97.22%	S	79,139	289,763	2066	20,510	391,478

* IMI treatment phenotypic results, sensitive (S), tolerant (T).

**Table 3 plants-13-01097-t003:** Primer sequence of four KASP markers for screening variations of *OsALS1*.

Marker	Primer Sequence (5′-3′)
ALS1^179V^	F1: *GAAGGTGACCAAGTTCATGCT*CTATGGGCGTCTCCTGGAAGA F2: *GAAGGTCGGAGTCAACGGATT*CTATGGGCGTCTCCTGGAAGG R: CCATCACGGGCCAGGTCCC
ALS1^548M^	F1: *GAAGGTGACCAAGTTCATGCT*CAACATTTGGGTATGGTGGTGCAAAT F2: *GAAGGTCGGAGTCAACGGATT*CAACATTTGGGTATGGTGGTGCAATG R: TCGCTCTCACATTCCGGGTTG
ALS1^627N^	F1: *GAAGGTGACCAAGTTCATGCT*ATCATGTCCTTGAATGCGCCCCCAT F2: *GAAGGTCGGAGTCAACGGATT*ATCATGTCCTTGAATGCGCCCCCAC R: CAGTCCGTGTAACAAAGAAGAGTGAAGTCC
ALS1^628E^	F1: *GAAGGTGACCAAGTTCATGCT*GTGCTGCCTATGATCCCAAGTGA F2: *GAAGGTCGGAGTCAACGGATT*GTGCTGCCTATGATCCCAAGTGG R: ACAGTCCTGCCATCACCATCC

Italic highlighted sequences are complementary sequence of FAM and HEM labeled primers stored in PCR mix, respectively. Terminal base underlined represent as the variations.

## Data Availability

The whole genomic sequence data used in this study were deposited in the Sequence Read Achieve (SRA) of the NCBI database (BioProject ID: PRJNA1077238).

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
