# Peer review of "Investigation of Imidazolinone Herbicide Resistance Gene with KASP Markers for Japonica/Geng Rice Varieties in the Huanghuaihai Region of China"

_plants, 2024, doi:10.3390/plants13081097_

Round 1

Reviewer 1 Report

Comments and Suggestions for Authors

The authors have presented information on the status of Imidazolinone herbicide resistance gene in rice varieties using KASP markers. It is an informative work.

Minor comments:

Line 43. I suggest changing “Weeds, a serious biotic stress that poses a threat to…” to “Weeds are a serious biotic stress that pose a threat to…”

Lines 156-157. “JG818 is the first IMI-tolerant rice variety to be introduced into…”. I suggest changing to “JG818 was the first IMI-tolerant rice variety introduced into…”.

Line 159. I suggest changing “…performed with JG818…” to “…performed on JG818”

Table 1. I suggest changing the Title to “Primer sequences of six markers closely linked to OsALS1

Lines 244-245. I suggest changing to “…has received much attention and has been favoured and spread…”

Line 361. I suggest changing “…following the following thermal cycling…” to “…using the following thermal cycling….”

Comments on the Quality of English Language

Minor editing

Author Response

Dear reviewer,

Thank you very much to provide the comments and to improve the quality of our manuscript. 

All the corresponding statements and grammatical errors in the manuscript have been corrected in the revised manuscript.

Best regards,

Zhaohui

Reviewer 2 Report

Comments and Suggestions for Authors

This study encompasses an analysis of 194 rice varieties in China, leading to the development of 4 KASP markers associated with IMI herbicide tolerance. Utilizing these KASP markers, the study successfully discerned 17 tolerance lines. The manuscript is well-crafted, encompassing not only marker development but also incorporating NGS and herbicide trials. Despite being conducted in potted conditions, the study effectively demonstrates the distinction between sensitive and tolerant strains.

I have several questions and suggestions:

1. Is there any agronomic data available, such as yield metrics?

2. Figure 1 should display fluorescent values on both the X and Y axes. Additionally, the model name of the PCR machine should be provided in section 4.5. Furthermore, the software used to present the figure should be specified.

3. In lines 186 and 187, the mutation positions 627N and 627S should be superscripted.

Author Response

Dear Reviewer,

We sincerely thank you for your valuable feedback that we have used to improve the quality of our manuscript. We have finished all corrections in revised version. The point-to-point response to your comments is attached as following. I look forward to that all the corrections will be met with approval.

Best regards,

Zhaohui

Point-to point response,

1. Is there any agronomic data available, such as yield metrics?

Response: The agronomic traits of rice varieties can be retrieved at the China Rice Data Center (https://ricedata.cn/variety/index.htm) or national big data platform for rice industry (http://202.127.42.47:6010/SDSite/Home/Index). The agronomic traits of regional tested lines are under investigation, we have not investigated yet.

2. Figure 1 should display fluorescent values on both the X and Y axes. Additionally, the model name of the PCR machine should be provided in section 4.5. Furthermore, the software used to present the figure should be specified.

Response: Thank you very much for great suggestion! The relative fluorescent values of both the X and Y axes are displayed in Figure1. We have highlighted the instrument models for PCR and KASP genotyping in section 4.3 and 4.5. Additionally, section 4.5 provides a description of how to use software for processing statistics and graphics.

3. In lines 186 and 187, the mutation positions 627N and 627S should be superscripted.

Response: Thank you so much for your careful check! We have corrected this formatting mistake.